# Interleukin-18 Is a Prognostic Biomarker Correlated with CD8^+^ T Cell and Natural Killer Cell Infiltration in Skin Cutaneous Melanoma

**DOI:** 10.3390/jcm8111993

**Published:** 2019-11-15

**Authors:** Minchan Gil, Kyung Eun Kim

**Affiliations:** 1Department of Stem Cell and Regenerative Biotechnology, Konkuk University, Seoul 05029, Korea; minchangil@gmail.com; 2Department of Cosmetic Sciences, Sookmyung Women’s University, Seoul 04310, Korea; 3Nano-Bio Resources Center, Sookmyung Women’s University, Seoul 04310, Korea

**Keywords:** Interleukin-18, melanoma, immune infiltrates, patient survival, multiomic analysis

## Abstract

Interleukin-18 (IL-18) is a cytokine that enhances innate and adaptive immune responses. Although there are conflicting reports about the roles of IL-18 in melanoma progression, the clinical relevance of IL-18 expression has not been comprehensively studied. In this study, we investigated IL-18 expression and its correlation with patient survival and immune cell infiltration in melanoma using cancer gene expression data publicly available through various databases. *IL18* mRNA expression was found to be significantly lower in melanoma tissues than normal tissues. Kaplan–Meier survival analysis showed that *IL18* expression was positively correlated with patient survival. To investigate the possible mechanisms by which *IL18* expression increased patient survival, we then assessed the correlation between *IL18* expression and immune cell infiltration levels. Infiltration of various immune cells, especially CD8^+^ T and natural killer (NK) cells, which are cytolytic effector cells, was significantly increased by *IL18* expression. Additionally, the expression levels of two cytolytic molecules including perforin and granzyme B were significantly positively correlated with *IL18* expression. Collectively, this study provides the first evidence that *IL18* expression has prognostic value for melanoma patient survival and is strongly correlated with CD8^+^ T and NK cell infiltration, suggesting the role of IL-18 as a biomarker for predicting melanoma prognosis.

## 1. Introduction

Interleukin-18 (IL-18) was first discovered as interferon-gamma (IFN-γ) inducing factor in the sera of mice injected with endotoxins [1]. It was eventually identified as IL-18 [2], and is currently classified as a member of the IL-1 superfamily cytokine because IL-18 is structurally related to IL-1, especially IL-1β. IL-18 is first synthesized as a 24 kDa precursor, which is then cleaved by caspase-1 into an 18 kDa bioactive form and secreted from various cell types including various immune and non-immune cells [3]. This cytokine is produced by activated immune cells, such as monocytes, macrophages, dendritic cells (DCs), neutrophils, natural killer (NK) cells, T cells, and B cells [2]. It stimulates IFN-γ production from T helper 1 (Th1) cells and enhances lymphocyte proliferation and so plays important roles in the regulation of innate and adaptive immunity [4]. As well as being a regulator of immune responses, IL-18 also participates in the immune escape of cancer cells, inflammatory responses, and autoimmune responses depending on the host environment [4,5]. Various types of cancer produce IL-18, and IL-18 induces cell migration, invasion, and proliferation, resulting in increased metastasis and tumor growth [5,6,7].

Melanoma, the most aggressive type of skin cancer, is characterized by highly metastatic tumors and, therefore, has a poor prognosis. Recently, the National Cancer Institute’s Surveillance, Epidemiology, and End Results (SEER) program found that melanoma is the fifth most common cancer in the United States [8]. Additionally, the incidence of melanoma is increasing faster than that of other types of cancer, leading to 287,723 new cases of melanoma diagnosed worldwide in 2018 [9]. Moreover, melanoma is resistant to various treatments, including chemotherapy, because it has a high rate of metastasis and resistance to apoptosis [10]. Although various therapies have been used to treat melanoma, none of them have significantly reduced melanoma mortality [11]. Melanoma cells have been shown to produce IL-18 spontaneously, which is related to the immune escape of melanoma through the enhancement of metastatic ability and tumor growth. Exogenous IL-18 directly enhances the migration and invasion of a melanoma cell line, B16F10 [12], suggesting that IL-18 has critical roles in melanoma malignancy.

Recently, the roles and prognostic value of tumor-infiltrating lymphocytes (TILs) have been extensively studied in various malignant tumors [13,14,15]. TILs are known to have critical effects on the survival of malignant melanoma, and act as prognosis markers [16]. Several studies have reported that high levels of lymphocyte infiltration lead to a better prognosis for melanoma patients, implying that TILs are a good prognostic factor [17,18]. TILs consist of multiple lymphocytes, such as CD4^+^ T cells, CD8^+^ T cells, B cells, NK cells, DCs, and macrophages [14]. The immune cells of this heterogeneous population have different effects on tumors depending on the tumor environment. CD8^+^ T cells, which comprise the majority of TILs, act as anti-tumor effector cells. Therefore, the substantial infiltration of CD8^+^ T cells is associated with a good prognosis in several tumors [16,19,20]. The other anti-tumor effector cells, NK cells are also found in tumors where they are regulated by various cytokines in the tumor environment, resulting in different functions [21]. DC-derived cytokines including IL-18, IL-12, and IL-15 activate NK cells, whereas transforming growth factor (TGF)-β and IL-10 from regulatory T cells or tumor cells inhibit NK cells [21]. Thus, studies of TIL populations and their roles in the tumor environment are needed for predicting prognosis.

Although there have been many conflicting reports about the roles of IL-18 in tumors, there has been no comprehensive analysis of the clinical relevance of IL-18 expression in melanoma. Therefore, in this study, we investigated the expression of IL-18 and its relevance on patient survival in melanoma. We systematically analyzed IL-18 expression and its correlation with patient survival using multiomic analysis tools. Additionally, to identify the associated factors affecting survival rates, we also investigated the correlation between IL-18 expression and the TILs in the tumor environment using the Tumor Immune Estimation Resource (TIMER). In conclusion, this study provides the first evidence for the potential of IL-18 expression as a biomarker for predicting melanoma prognosis and its correlation with infiltration by cytotoxic CD8^+^ T and NK cells.

## 2. Experimental Section

### 2.1. Comparison of *Interleukin-18 (IL18)* mRNA Expression in Various Types of Tumors and Their Normal Tissue Counterparts

*IL18* mRNA expression in various cancers and their normal tissue counterparts were analyzed using the Gene Expression Profiling Analysis (GEPIA) (Beijing, China) [22,23] and Gene Expression across Normal and Tumor tissue (GENT) databases (Korea Research Institute of Bioscience and Biotechnology, Daejeon, Korea) [24,25]. GEPIA provides RNA sequencing data from of the Cancer Genome Atlas (TCGA) of tumor samples with paired adjacent TCGA and Genotype-Tissue Expression (GTEx) normal tissue samples. TCGA and GTEx RNA-Seq expression datasets in GEPIA are based on the UCSC (University of California, Santa Cruz) Xena project [26], which are recomputed based on a uniform bioinformatic pipeline to eliminate batch effects. To compare expression data, data are normalized by quantile-normalization [27] or other two additional normalization strategies [22]. The GENT database provides gene expression data across various human cancer and normal tissues profiled with the Affymetrix U133A or U133plus2 platforms. Data were collected from public resources, processed by MAS5 algorithm using the affy package [28] and normalized target density 500 [24]. All queries of both databases were performed with defaults settings. *IL18* expression in normal and melanoma samples from the Oncomine database version 4.5 (Thermo Fisher Scientific Inc., Ann Arbor, MI, USA) were also explored with threshold *P*-value < 0.05 and fold change > 2 [29,30,31].

### 2.2. Analysis of Correlation between *IL18* mRNA Expression and Patient Survival in Various Tumors

The correlation between *IL18* mRNA expression and patient survival in the TCGA data was evaluated using the OncoLnc (A site by Jordan Anaya, Berkeley, CA, USA) online analysis tool [32,33]. The correlation between *IL18* expression and overall patient survival in the TCGA data was also estimated using GEPIA. Patient cases were divided into two groups: high *IL18* TPM group, which includes half of cases with higher *IL18* expression above the median *IL18* expression level among cases and low *IL18* TPM group which includes another half case. The correlation of survival and gene expression was compared between two groups using Kaplan–Meier survival curves and the log-rank test using GEPIA. The *P*-value cut-off was set as 0.05. The number of patients with each clinicopathological characteristic in TCGA subsets were shown at previous published article [34]. *IL18* expression in high and low risk groups were compared with box plot using the SurvExpress biomarker validation tool version 2.0 (Monterrey, Nuevo Leon, Mexico) [35,36]. The risk groups were split by the median prognostic index (PI). Kaplan Meier Scanner from the R2 version 3.2.0 (Department of Oncogenomics of the Academic Medical Center, Amsterdam, the Netherlands) [37] was used to generate survival curves to compare the two patient groups split by the level of *IL18* expression. The cutoff value for the groups was selected to minimize the log-rank *P*-value.

### 2.3. Analysis of *IL18* Gene Mutations and Copy Number Alterations (CNA) in Skin Cutaneous Melanoma (SKCM)

Mutation and CNA analyses were conducted on the TGCA PanCanAtlas datasets using the cBioPortal database version 2.2.0 (Center for Molecular Oncology at MSK, New York, NY, USA) [38,39,40]. The mutation diagram and alteration frequency of the *IL18* gene were generated with the default parameter settings. Somatic copy number alterations were determined with the Genomic Identification of Significant Targets in Cancer (GISTIC) algorithm. *IL18* expression was examined for each alteration status (deep deletion, shallow deletion, diploid, and gain) and plotted. The unpaired *t*-test from GraphPad 7 software was used for statistical analysis.

### 2.4. Analysis of the Correlation between *IL18* Expression and the Immune Cell Infiltration

The correlation between *IL18* expression and the abundance of infiltrating immune cells in the TCGA datasets was investigated with the Tumor Immune Estimation Resource (TIMER) web tool (X Shirley Liu Lab & Jun Liu Lab at Harvard university, Boston, MA, USA) [41,42]. The correlation of *IL18* expression level with tumor purity and the abundance of B cells, CD4^+^ T cells, CD8^+^ T cells, macrophages, neutrophils, and dendritic cells were displayed for each tumor. The correlation between *IL18* expression and the gene markers of immune cell subsets were explored via the correlation modules in the TIMER web tool and the Spearman’s correlation and the estimated statistical significance were calculated. The gene markers for each type of infiltrating immune cells were chosen as previously reported with a slight modification [43]. GEPIA was used to confirm the significance of the correlation between *IL18* and the marker genes for infiltrating immune cells.

## 3. Results

### 3.1. *IL18* mRNA Expression Levels in Various Types of Cancer

To compare the levels of *IL18* expression in tumor and normal tissues, the *IL18* mRNA levels in various types of cancer and their normal tissue counterparts were analyzed using the RNA-seq data from the TCGA and GTEx datasets. As shown in Figure 1a, compared with normal tissues, the *IL18* transcription levels were significantly lower in ACC (adrenocortical carcinomas), ESCA (esophageal carcinoma), and SKCM (skin cutaneous melanoma) tumor tissues but was increased in most cancer types, including CESC (cervical squamous cell carcinoma and endocervical adenocarcinoma), COAD (colon adenocarcinoma), DLBC (lymphoid neoplasm diffuse large B-cell lymphoma), GBM (glioblastoma), KICH (kidney chromophobe), KIRP (kidney renal papillary cell carcinoma), LAML (acute myeloid leukemia), LGG (brain lower grade glioma), OV (ovarian serous cystadenocarcinoma), PAAD (pancreatic adenocarcinoma), READ (rectum adenocarcinoma), STAD (stomach adenocarcinoma), TGCT (testicular germ cell tumors), THCA (thyroid adenocarcinoma), THYM (thymoma), and UCEC (uterine corpus endometrial carcinoma). Additionally, to confirm the differences in *IL18* expression in tumor and normal tissues, *IL18* expression was analyzed using the GENT database. Figure 1b shows that *IL18* mRNA expression was lower in skin cancer, colon cancer, and lung cancer, however, higher expression was detected in most other types of cancer, including breast cancer, brain cancer, and pancreatic cancer. Collectively, most tumor tissues had higher *IL18* mRNA expression than normal tissues; however, interestingly, SKCM had lower expression.

### 3.2. Correlation between *IL18* Expression and Patient Survival Rates in Various Types of Cancers

To investigate whether the cancer patient survival rate is correlated with *IL18* expression levels in various types of cancers, the overall survival rates of the 50% of patients with higher levels of *IL18* expression and the 50% with lower levels of *IL18* expression were compared with the Cox regression model using the OncoLnc online tool. Cox regression results for *IL18* in various types of cancer are shown in Appendix A. Significant Cox regressions for *IL18* expression were observed in five types of cancer, SKCM, SARC (sarcoma), BRCA (breast invasive carcinoma), LGG, and PAAD (*P* < 0.05). The *IL18* expression level was significantly positively correlated with patient survival rate in SKCM, SARC, and BRCA, but was negatively correlated with patient survival in LGG and PAAD. There was no significant correlation between *IL18* expression and the survival rates in colon adenocarcinoma (COAD), which was used as a negative control in subsequent analysis. Additionally, overall survival of these six types of cancer were visualized using Kaplan–Meier survival curves. As shown in Figure 2a–c, high *IL18* expression was associated with longer survival in SKCM, SARC, and BRCA, leading to a good prognosis (Log-rank *P* < 0.05). In contrast, poor prognosis in two types of cancer, LGG and PAAD, were correlated with high *IL18* expression (Log-rank *P* < 0.05, Figure 2d,e). Data for COAD as a negative control showed no correlation between *IL18* expression and prognosis (Figure 2f). Taken together, these results suggest that *IL18* mRNA expression has different prognostic effects in different cancer types. Of note, a significant positive correlation between *IL18* expression and survival rates was found in SKCM, which has lower *IL18* expression than normal tissues, implying that the malignancy of melanoma is due to the lower *IL18* expression.

### 3.3. *IL18* Expression Pattern and Patient Survival in SKCM

To further examine the *IL18* expression in SKCM compared to normal tissues, we analyzed other datasets in the Oncomine database. Relative *IL18* expression was significantly downregulated in melanoma tissue compared to normal tissue in the Riker and Talantoy melanoma datasets (*P* = 7.80 × 10^−4^ and 0.021, respectively, Figure 3a and Appendix A). To analyze the association between *IL18* expression and patient risk, we determined the *IL18* expression in high- and low-risk groups with the SurvExpress biomarker validation tool. As shown in Figure 3b, the *IL18* mRNA was significantly lower in the high-risk group (*n* = 22) than the low-risk group (*n* = 22) in the GEO dataset GSE19234. In addition, we compared patient survival between the high and low *IL18* expression groups using R2: Kaplan–Mayer Scanner. In the Tumor Melanoma Metastatic-Bhardwaj-44 dataset, patients with high *IL18* expression (*n* = 16) had significantly higher overall survival than patients with lower *IL18* expression (*n* = 28) (Figure 3c). Collectively, these data-driven results suggest that *IL18* expression is significantly downregulated in melanoma cells and is positively correlated with patient survival. Next, *IL18* gene alterations in the TCGA PanCan Altas dataset were analyzed using the cBioPortal website. Five missense mutations were found in the *IL18* coding region. The E192K mutation was found 3 times more frequently than the others (Figure 3d). In total, 4.69% of *IL18* genes were altered, consisting of 1.38% mutations, 3.31% deep deletions, and 3.31% of mRNA high (Figure 3e). To determine whether the *IL18* copy number alteration (CNA) status correlated with mRNA expression, we determined *IL18* mRNA expression for each CNA status. Figure 3f shows that *IL18* expression was significantly lower in the deep-deletion samples than the shallow deletion and diploid samples. However, there were no significant differences in *IL18* expression between gain samples and any other CNA status samples. These data implied deep deletion CNA status could partially contribute that lower expression of *IL18* in melanoma.

### 3.4. Correlation of *IL18* Expression with Immune Infiltrates in SKCM

To investigate the mechanisms and relationship between *IL18* and good prognosis in melanoma, we focused on immune cell infiltrates in the melanoma tumor environment. It has been reported that high levels of lymphocyte infiltration lead to a better prognosis in melanoma patients, implying that TILs act as a good prognostic factor [17,18]. To determine the role of the *IL18* expression in TIL abundance, TIMER was used to analyze the correlation between *IL18* expression and the infiltration levels of various immune cells. Figure 4a shows that *IL18* expression was significantly negatively correlated with tumor purity in SKCM (cor. = −0.674, *P* = 5.73 × 10^−62^), indicating that *IL18* expression in tumor tissues might be expressed by the infiltrating immune cells. Higher *IL18* expression in SKCM tissues markedly increased the infiltration of immune cells, such as B cells (cor. = 0.202, *P* = 1.69 × 10^−5^), CD8^+^ T cells (cor. = 0.464, *P* = 9.90 × 10^−25^), CD4^+^ T cells (cor. = 0.265, *P* = 1.33 × 10^−8^), macrophages (cor. = 0.384, *P* = 2.36 × 10^−17^), neutrophils (cor. = 0.506, *P* = 9.72 × 10^−21^), and dendritic cells (cor. = 0.533, *P* = 4.60 × 10^−34^). It is well known that various immune cells including monocytes, macrophages, and dendritic cells produce IL-18 [2]. Therefore, the infiltrating immune cells seem to be a major source of IL-18 in the melanoma tumor environment.

Additionally, there was also a negative correlation with tumor purity and the infiltration of most immune cells increased along with *IL18* expression in SARC and BRCA. The other cancer types showed a positive correlation between *IL18* expression and patient survival rates similar to that seen with SKCM as shown in Figure 4b and c. In SARC, a significant correlation was shown in most immune cells (B cells; cor. = 0.333, *P* = 1.34 × 10^−7^, CD8^+^ T cells; cor. = 0.239, *P* = 1.84 × 10^−4^, CD4^+^ T cells; cor. = 0.684, *P* = 2.55 × 10^−34^, macrophages; cor. = 0.632, *P* = 1.21 × 10^−27^, neutrophils; cor. = 0.494, *P* = 2.92 × 10^−16^, dendritic cells; cor. = 0.708, *P* = 5.36 × 10^−38^). Also, *IL18* expression levels were positively correlated with immune infiltrates in BRCA (B cells; cor. = 0.367, *P* = 1.55 × 10^−32^, CD8^+^ T cells; cor. = 0.217, *P* = 6.50 × 10^−12^, CD4^+^ T cells; cor. = 0.369, *P* = 2.28 × 10^−32^, macrophages; cor. = 0.131, *P* = 3.87 × 10^−5^, neutrophils; cor. = 0.39, *P* = 5.40 × 10^−36^, dendritic cells; cor. = 0.44, *P* = 2.42 × 10^−46^). However, *IL18* expression did not have any correlation with tumor purity or immune infiltration in COAD which was used as a negative control (Figure 4d). These results demonstrate that there is a positive correlation between *IL18* expression and immune cell infiltration levels in cancer types, such as SKCM, SARC, and BRCA in which *IL18* expression level is positively correlated with good prognosis.

### 3.5. Correlation between *IL18* Expression and Various Subsets of Immune Cells in Melanoma

To further determine the correlation between *IL18* expression levels and various subsets of immune cell infiltrations in SKCM, we analyzed the correlations between *IL18* and immune cell markers including subsets of T cells, B cells, monocytes, M1 and M2 macrophages, neutrophils, NK cells, and dendritic cells (DCs) in SKCM as shown in Table 1, Table 2 and Figure 5. Table 1 and Appendix A show that *IL18* expression was significantly correlated with most immune cell markers in SKCM, SARC, and BRCA, whereas *IL18* was not significantly correlated with gene markers of various immune cells in COAD. Only 20 gene markers in COAD showed significant correlations which were considerably lower than those seen in SKCM. Of these immune cells, CD8^+^ T cells comprise the majority of TILs and directly induce cell death in tumors, thus, a large number of CD8^+^ T cells in the tumor environment might be associated with a favorable prognosis [16,19,20]. It is also well known that NK cells have spontaneous killing activities against tumors as anti-tumor effecting cells [21]. Additionally, γδ T cells could be a better prognostic indicator in cancer than CD8^+^ T cells and NK cells [44].

Here, infiltration of anti-tumor effector cells, CD8^+^ T cells, NK cells, and γδ T cells was markedly increased by *IL18* expression (Table 1, Appendix A). To further confirm this, the GEPIA database was used to analyze the correlation between *IL18* expression level and gene markers of CD8^+^ T cells, NK cells, and γδ T cells (Table 2, Appendix A). To identify γδ T cell-related genes, we used transcriptomic data of two genes (*CCR5* and *CXCR6*) from the LM22 gene set of CIBERSORT along with *RORC* and *CD27*. Newman AM et al. reported significantly differentially expressed genes in γδ T cells by comparing each 22 leukocyte subsets [45]. There was significant correlation between *IL18* expression and markers of CD8^+^ T cells, NK cells, and γδ T cells in SKCM, SARC, and BRCA but not in COAD. Interestingly, there were no significant positive correlations in normal tissue counterparts of SKCM, however, *IL18* expression was positively correlated with CD8^+^ T cell, NK cell, and γδ T cell infiltration in the normal tissue counterparts of COAD (Table 2). Additionally, there is also strong positive correlation between *IL18* expression and expression of γδ T cells-related genes in SKCM as shown in Appendix A, implying the possibility of γδ T cell infiltration in SKCM by *IL18* expression. Collectively, these data indicate that higher *IL18* expression is significantly involved in the infiltration of CD8^+^ T cells, NK cells, and γδ T cells in SKCM, resulting in a higher survival rate via the anti-tumor activities of the effector cells.

### 3.6. Correlation between *IL18* Expression and Gene Expression of Cytolytic Molecules, Granzymeb and Perforin

It is well known that CD8^+^ T cells and NK cells are representative cytolytic effector cells which lyse transformed, virus-infected, and tumor cells. NK cells have spontaneous cytolytic activities, and their activities are enhanced by stimulation with specific cytokines like IL-2, IL-12, IL-15, and IL-18 [46]. The cytolytic activities of CD8^+^ T cells and NK cells are commonly mediated by the release of cytotoxic molecules such as granzyme B and perforin. Perforin creates pores on the surface of target cells while granzyme B enters through the pores and then activates apoptosis signaling in the target cells, leading to cell death [47,48,49]. Here, to investigate whether *IL18* expression is associated with the gene expression of cytolytic molecules in the infiltrating CD8^+^ T and NK cells, we analyzed the correlation between *IL18* expression and gene expression of the two cytolytic molecules, *GZMB* (granzyme B) and *PRF1* (perforin), using the TIMER database. Figure 6 shows that *IL18* expression was dramatically positively correlated with *GZMB* (cor. = 0.688, *P* = 2.71 × 10^−67^) and *PRF1* (cor. = 0.7, *P* = 0.00 × 10^0^) gene expression in SKCM (*n* = 103). In addition to SKCM, *IL18* expression is also positively correlated with *GZMB* (cor. = 0.52, *P* = 3.03 × 10^−77^) and *PRF1* (cor. = 0.55, *P* = 0.00 × 10^0^) gene expression in BRCA (*n* = 1093) and *GZMB* (cor. = 0.668, *P* = 0.00 × 10^0^) and *PRF1* (cor. = 0.732, *P* = 0.00 × 10^0^) expression in SARC (*n* = 259). However, there was no correlation between *IL18* expression and those two cytolytic molecules in COAD (*GZMB*; cor. = 0.061, *P* = 1.93 × 10^−1^, *PRF1*; cor. = 0.067, *P* = 1.54 × 10^−1^) as shown in Figure 6. These data suggest that the infiltrating CD8^+^ T cells and NK cells could inhibit tumor progression via the release of granzyme B and perforin, resulting in the improvement in patient survival.

## 4. Discussion

IL-18 is a representative proinflammatory cytokine and it is known to have both anti-cancer and pro-cancer properties depending on the host environment [4,5]. Several studies have reported that tumor progression and IL-18 levels are positively correlated in various types of cancer [50,51,52,53]. Serum IL-18 is increased in patients with cancer, and high levels of serum IL-18 is correlated with poor clinical outcomes in some types of cancers including breast and pancreatic cancers [50,52]. In vitro studies show the direct effects of exogenous IL-18 on melanoma [12,54]. Exogenous IL-18 increases cell migration in the melanoma cell line B16F10 [12], and cell adhesion in both murine and human melanoma cell lines [54], implying that IL-18 enhances malignant properties in melanoma cell. On the other hand, a tumor suppressing effect of IL-18 has been suggested due to its ability to activate immune cells, such as NK cells and T cells [55]. IL-18 directly enhances NK cell proliferation and migration to secondary lymphoid organs via upregulation of chemokine receptors, allowing them to inhibit the lymphoid spread of tumor cells [56]. In addition to the direct effects of IL-18 on NK cells, the expression of UL16 binding protein 2, which is a ligand for the NK cell-activating receptor, on tumor cells is increased by exogenous IL-18, leading to the increased NK cell cytotoxicity [57]. However, the mechanisms for anti-cancer or pro-cancer effects of IL-18 have not been clearly understood yet.

In this regard, a lot of studies have suggested two reasons, such as combination effects with other cytokines and the dose effect of IL-18. The pro-cancer effects of IL-18 in melanoma mentioned above have been studied in vitro, and these in vitro studies usually cannot reflect the actual tumor microenvironment in which various other regulatory factors could regulate cancer cells synergistically or antagonistically with IL-18. A lot of studies have suggested that IL-18 shows anti-cancer effects in combination with other cytokines including IL-12 and IL-15 as results of inducing T cell proliferation and NK cell activation [58,59] in vitro. Indeed, Mirjačić Martinović K et al. reported that NK cell cytotoxicity was increased in IL-18/IL-12 treated NK cells isolated from patients with malignant melanoma, whereas IL-18 alone did not show those effect [60]. Müller J et al. reported that IL-18/IL-12 combination gene therapy showed significant tumor regression in a malignant melanoma-induced horse model [61]. Also, overexpression of IL-18 in a mouse model of malignant melanoma has been shown to increase tumor cell apoptosis, inhibit tumor growth, reduce lung metastasis, and inhibit angiogenesis, implying that IL-18 shows systemically a significant anti-cancer effect in the body [62]. Next, several studies have shown that IL-18 has anti-cancer or pro-cancer effects depending on the doses. In particular, the anti-cancer effects of IL-18 accompanied by the activation of NK cells and T cells are often manifested in high doses of IL-18 in combination with IL-2 or IL-12 [63,64,65]. Actually, it is well known that the effectiveness of cytokine in cancer therapy is significantly dependent on cytokine doses and schedules [66]. These anti-cancer effects of high-dose IL-18 is consistent with our data, which indicate that higher expression of IL-18 has a better prognosis. Collectively, IL-18 might show both anti-cancer and pro-cancer effects depending on tumor environment in the body [4]. Therefore, further investigations are needed about the contradictory findings related to the biological role of IL-18 in vitro and in vivo.

As such, the role of IL-18 in tumor progression remains controversial, and there is no comprehensive analysis of IL-18 expression and its correlation with clinical outcomes in patients with melanoma. Therefore, a systematic analysis of IL-18 expression and the correlation between IL-18 and patient survival in melanoma is definitely needed in order to comprehensively understand the roles of IL-18 in the body of melanoma patients. Our study found that *IL18* mRNA expression in melanoma tumor tissue was lower than that in normal tissues and higher *IL18* expression leads to increased survival rates (Figure 1 and Figure 2). These data imply that poor clinical outcomes of melanoma patients would be due to lower *IL18* expression. Furthermore, our analyses show that the *IL18* expression level is positively correlated with the level of infiltration by various immune cells, including B cells, CD8^+^ T cells, CD4^+^ T cells, macrophages, neutrophils, and dendritic cells (Figure 4). Several studies have reported that high levels of lymphocyte infiltration lead to a better prognosis in melanoma patients, implying that TILs are a good prognostic factor [17,18]. In particular, Gentle AJ et al. suggested that γδ T cells, CD8^+^ T cells, and NK cells are important indicators of favorable outcomes in pan-cancer and solid cancer using the CIBERSORT web tool [44]. As shown in Figure 4, the infiltration by various immune cells was strongly correlated with *IL18* expression. It is also well known that IL-18 is produced by various immune cells, such as macrophages, DCs, neutrophils, NK cells, CD4^+^ T cells, CD8^+^ T cells, γδ T cells, and B cells [2,67]. Because tumor purity shows a strong negative correlation with *IL18* expression in SKCM as shown in Figure 4, it is likely that the source of the IL-18 is the infiltrated immune cells [41].

Notably, the strongest positive correlation was shown between the expression of specific markers for CD8^+^ T cells, NK cells, and γδ T cells and *IL18* expression (Table 1 and Figure 5). CD8^+^ T cells, NK cells, and γδ T cells are well known as effector cells in the tumor environment via their cytolytic activity [68,69,70,71,72]. These effector cells have cytolytic activity through the production of cytolytic molecules, such as perforin and granzyme, the Fas/Fas ligand interaction, and NKG2D/NKG2D ligand interaction leading to cancer cell apoptosis [70,72,73]. In fact, this study shows a positive correlation between *IL18* expression and gene expression of two cytolytic molecules, such as *GZMB* (granzyme B) and *PRF1* (perforin), and various effector cells activating markers expressed on the effector cells, such as *KLRK1* (NKG2D), *NCR1* (NKp46), *NCR2* (NKp44), and *NCR3* (NKp30) in SKCM as shown in Figure 6, suggesting that IL-18 could increase patient survival due to the enhanced infiltration and cytolytic activities of CD8^+^ T cells, NK cells, and γδ T cells. Also, it was confirmed that those genes are expressed in the infiltrated effector cells through the analysis of tumor purity in the TIMER web tool (data not shown). Consistent with our analysis, previous studies using CIBERSORT web tool also have reported that higher levels of CD8^+^ T cells and NK cells in SKCM are positively correlated with a better prognosis, suggesting that the increased infiltration of those effector cells is an important target for predicting prognosis of cancer patients with SKCM [74,75].

As such, this study provided the first clinical significance that *IL18* expression could be a good prognostic factor in patients with SKCM via inducing immune cell infiltration; however, there are some limitations. As shown in Appendix A, we selected four genes (*RORC, CD27, CCR5*, and *CXCR6*) to distinguish γδ T cells referring to previous study [45,76]. During the T cell development process, various types of γδ T cells including IL-17-producung or IFN-γ producing innate-like γδ T cells or adaptive γδ T cells are generated [76]. For example, γδ T cells with strong TCR signal drive cells into IFN-γ producing innate-like γδ T cells, and these cells express T-bet as a transcription factor and CD27 as a surface molecule. By contrast, IL-17-producing innate-like γδ T cells are generated upon weak signal, and the cells express RORγt as a transcription factor [76]. In addition, gene expression of CCR5 and CXCR6 is also used to identify γδ T cells. Newman AM et al. provided a list of significantly differentially expressed genes within 22 leukocyte subsets [45]. Based on these research works, the correlation of *IL18* expression with expression of γδ T cells-related four genes was analyzed, and there is also positive correlation in SKCM, SARC, and BRCA, but not COAD. Interestingly, *RORC* (RORγt) gene expression was shown to be low or no correlation with *IL18* expression, implying that IFN-γ producing innate-like γδ T cells could have more critical roles in *IL18*-regulated immune infiltration. Although γδ T cells act as important effector cells for better prognosis in cancer patients, and this study shows the positive correlation between *IL18* and γδ T cells-related genes, it is well known that those genes are also expressed in other effector cells, CD8^+^ T cells and NK cells [77,78]. Therefore, we cannot confirm the association between *IL18* expression and γδ T cells’ infiltration despite this study analyzed the correlation using γδ T cells-related genes with references [45,76].

Additionally, this study shows that *IL18* expression also had a positive correlation with *PD1* and *CTLA4* genes, which are known as immune suppressive molecules as shown in Table 1. Several studies have shown that IL-18 increases PD-1 expression, leading to immunosuppression in NK cells in melanoma and breast cancer [79,80]. Recently, anti-PD-1 or anti-PD-L1 therapies have been successful in treating certain cancers, which expressed PD-L1; however, its effectiveness is limited by the infiltration of immune cells and the characteristics of cancer [81,82]. Despite the increased expression of immune suppressive molecules, this study found that SKCM patients with higher *IL18* expression showed overall higher survival rates. Therefore, we suggest the need for additional analysis to confirm the efficacy of anti-PD-1 therapeutics according to differences in *IL18* expression.

## 5. Conclusions

In conclusion, this study shows that higher *IL18* mRNA expression correlates with the increased patient survival and infiltration of T cells, B cells, macrophages, neutrophils, and DCs in SKCM. Infiltration levels of effector cells including CD8^+^ T cells, NK cells, and γδ T cells, as shown by two cytolytic molecules produced by these effector cells, were markedly increased along with *IL18* mRNA expression. Therefore, our systematic analysis provides an integrated understanding of the potential role of IL-18 in melanoma and its use as a useful biomarker for the prognosis of melanoma patients.

## Figures and Tables

**Figure 1 jcm-08-01993-f001:**
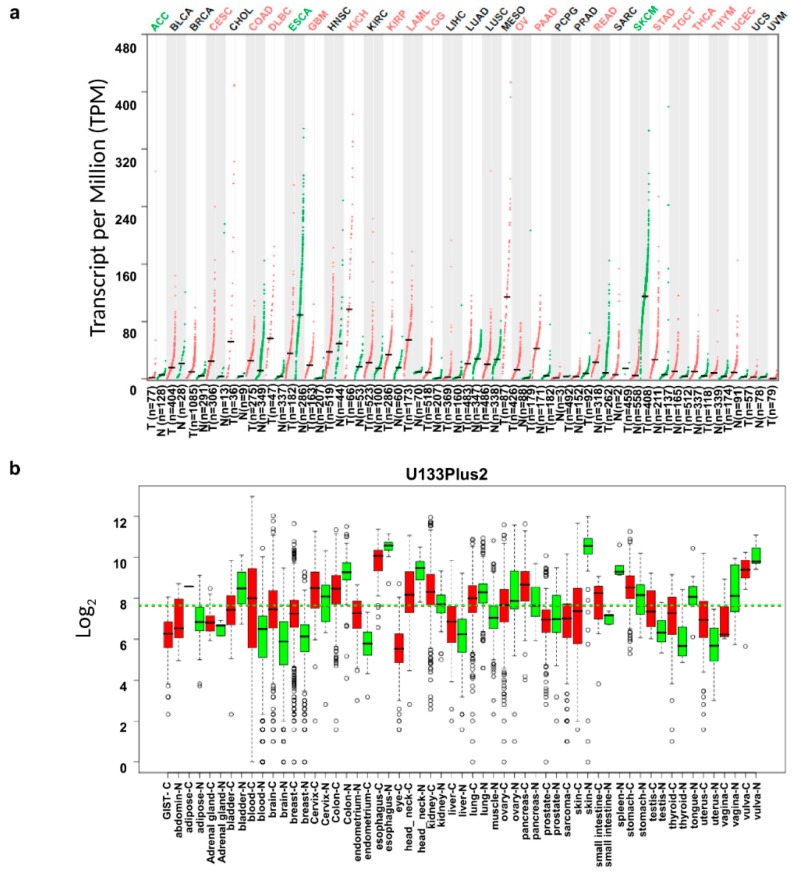
Transcription levels of *interleukin-18* (*IL18*) in different types of cancer and normal tissues. (**a**) Expression of *IL18* in various types of cancer and normal tissues visualized as dot plots based on the RNA-seq data from the Cancer Genome Atlas (TCGA) and Genotype-Tissue Expression (GTEx) datasets through the Gene Expression Profiling Interactive Analysis (GEPIA) website (http://gepia.cancer-pku.cn/). Abbreviations of cancers types are presented in Appendix A. Cancer types in which *IL18* expression was significantly higher or lower than normal tissues are marked as red or green, respectively. (**b**) *IL18* mRNA expression in various types of cancer from the Gene Expression across Normal and Tumor tissue (GENT) database [25]. The boxplots show mRNA expression data from each cancer and normal tissue profiled by the U133plus platforms. Boxes show the median and the 25th and 75th percentiles; dots are outliers. Red boxes represent tumor tissues; green boxes represent normal tissues. Red and green dashed lines represent the average value of all tumor and normal tissues, respectively.

**Figure 2 jcm-08-01993-f002:**
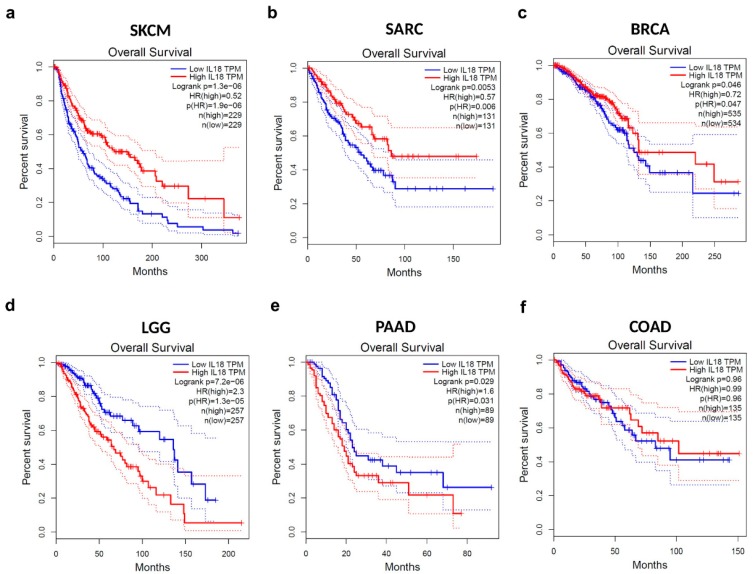
Correlation between *IL18* expression and overall patient survival in various types of cancers. Kaplan–Meier survival curves with higher (red) and lower (blue) *IL18* expression than the median were generated from TCGA data using the GEPIA website for (**a**) skin cutaneous melanoma (SKCM; *n* = 458), (**b**) sarcoma (SARC; *n* = 262), (**c**) breast invasive carcinoma (BRCA; *n* = 1069), (**d**) brain lower grade glioma (LGG; *n* = 514), (**e**) pancreatic adenocarcinoma (PAAD; *n* = 178), and (**f**) colon adenocarcinoma (COAD; *n* = 270). The log-rank *P*-value, Cox proportional hazard ratio (HR), *P*-value of HR, and number of patients in each analysis group are included. The 95% confidence interval is plotted with a dotted line. TPM—transcript per million.

**Figure 3 jcm-08-01993-f003:**
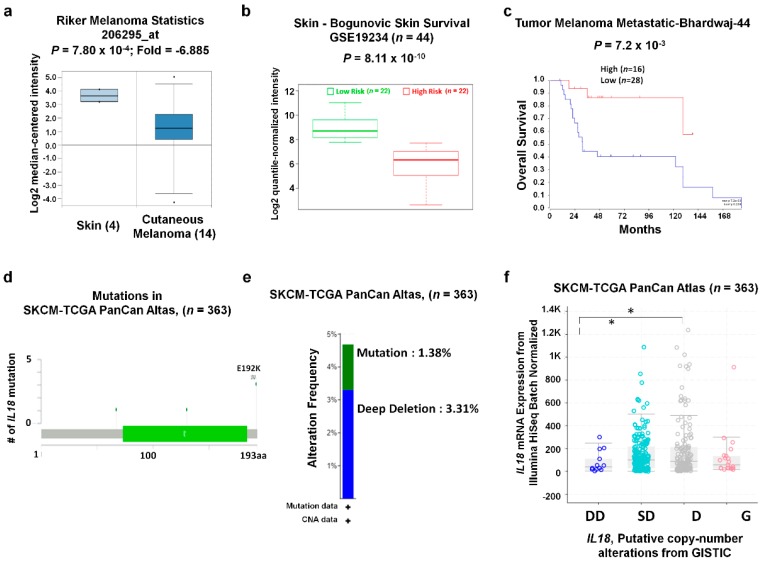
*IL18* expression and genome alterations in SKCM. (**a**) A box plot comparing specific *IL18* expression in normal (*n* = 4, left plot) and cancer tissues (*n* = 14, right plot) was derived from the Riker Melanoma dataset in the Oncomine database. (**b**) *IL18* expression in high (red) and low (green) risk patient groups in the Gene Expression Omnibus (GEO) dataset GSE19234 were visualized with a box-plot using the SurvExpress biomarker validation tool [36]. Expression of *IL18* was represented quantile-normalized and log2 transformed intensity data in GSE19234 dataset in *Y*-axis. The *t*-test was used to compare the expression of the high and low risk groups. (**c**) The survival curves comparing the patient groups with high (red) and low (blue) *IL18* expression in the Tumor Melanoma Metastatic-Bhardwaj-44 dataset was plotted using R2: Kaplan–Meier Scanner [37]. Survival curve analysis was conducted using a threshold Cox *P*-value < 0.05. (**d**) A schematic diagram shows location and frequency of each mutation across the *IL18* coding sequence in SKCM-TCGA PanCan Altas dataset) (*n* = 363). obtained from cBioPortal database (http://www.cbioportal.org). The *IL18* mutations mainly occurred in one hot spot (E192K). (**e**) Frequencies of mutation and copy number alteration (CNA) of *IL18* in SKCM -TCGA, PanCan atlas dataset (*n* = 363) determined using the cBioPortal website. (**f**) *IL18* mRNA expression was not significantly associated with CNA status: deep deletion (DD), shallow deletion (SD), diploid (D), gain (G). GEO datasets used in Figure 3a–c were listed in Appendix A.

**Figure 4 jcm-08-01993-f004:**
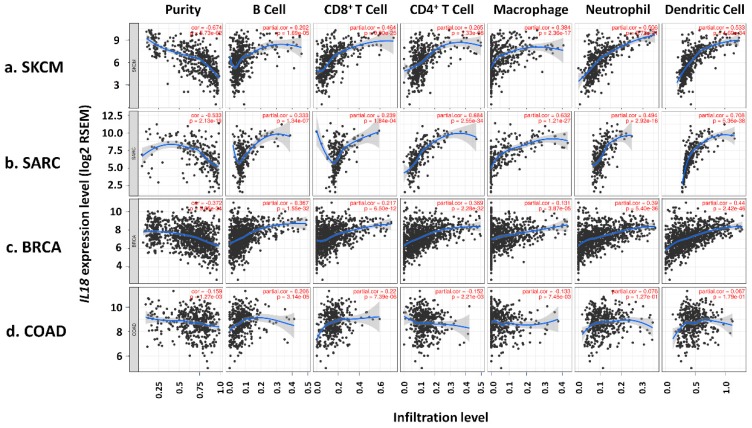
Correlation of *IL18* expression with immune infiltration in SKCM, SARC, BRCA, and COAD. A correlation between *IL18* expression and the abundance of infiltrating immune cells in the TCGA datasets was investigated with the Tumor Immune Estimation Resource (TIMER) web tool [42] (**a**) *IL18* expression is significantly negatively correlated with tumor purity and positively correlated with levels of various infiltrating immune cells including B cells, CD8^+^ T cells, CD4^+^ T cells, macrophages, neutrophils, and dendritic cells. Similar to SKCM, *IL18* expression and immune infiltration were positively correlated in (**b**) SARC and (**c**) BRCA. (**d**) *IL18* expression was not significantly correlated with tumor purity and infiltrating levels of immune cells in COAD. Correlation constants and *P*-values are listed in Appendix A.

**Figure 5 jcm-08-01993-f005:**
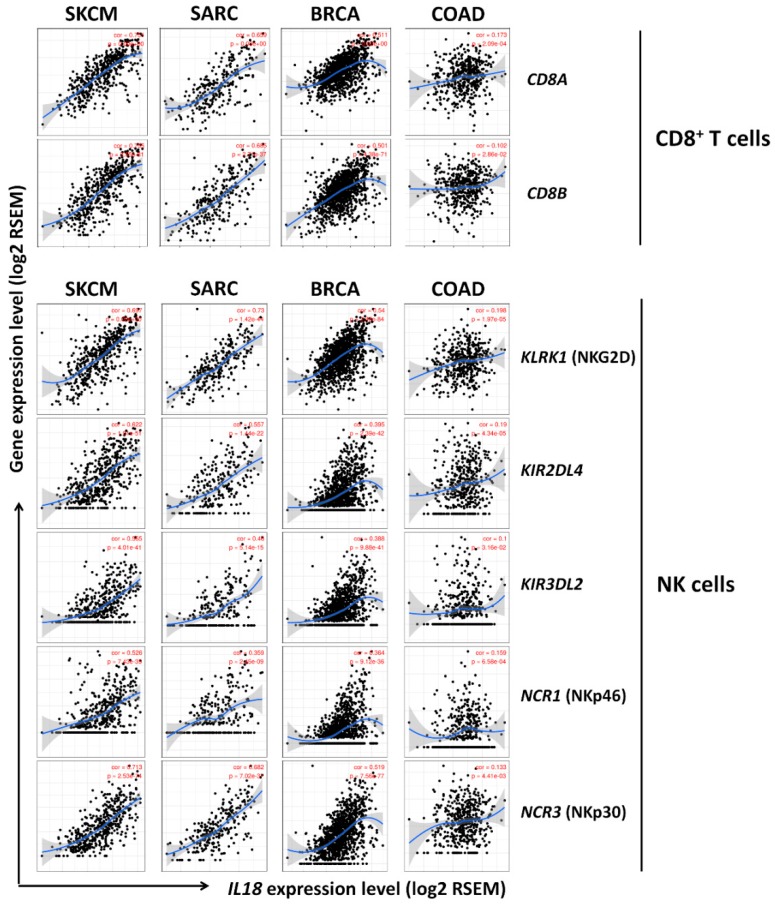
Correlation of *IL18* expression to markers of CD8^+^ T and NK cells in SKCM. Correlation of *IL18* expression with various gene markers of CD8^+^ T and NK cells was examined with the TIMER website. *CD8A* and *CD8B* genes were used as markers for CD8^+^ T cells, and *KLRK1, KIR2DL4, KIR3DL2, NCR1,* and *NCR3* were used as markers for NK cells. *IL18* expression was positively correlated with the expression of various gene markers for CD8^+^ T and NK cells in SKCM (*n* = 103), SARC (*n* = 259) and BRCA (*n* = 1093). There are no significant correlations between *IL18* expression and the expression of most gene markers in COAD (*n* = 457). Only the expressions of one gene marker, *KLRK1* was weakly correlated with *IL18* expression in COAD. Correlation constants and *P*-values are listed in Table 1.

**Figure 6 jcm-08-01993-f006:**
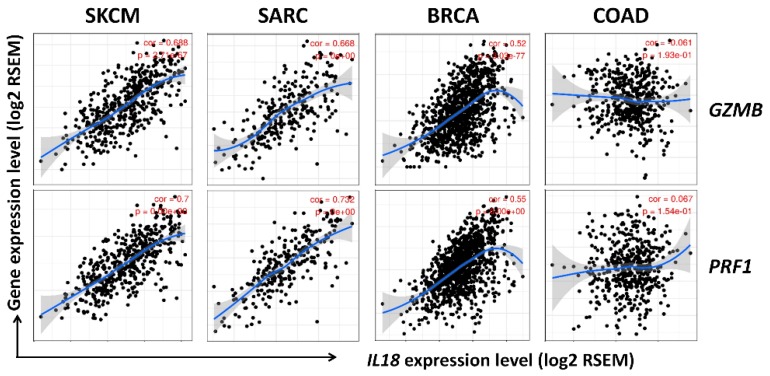
Correlation of *IL18* expression with granzyme B (*GZMB*) and perforin (*PRF1*) in SKCM. Correlations of *IL18* expression with the expression of two cytolytic molecule gene markers (*GZMB* and *PRF1*) were examined using the TIMER website. A strong positive correlation between *IL18* expression and both cytolytic molecule genes were seen in SKCM, BRCA, and SARC. *IL18* expression was not correlated with either gene in COAD.

**Table 1 jcm-08-01993-t001:** Correlation between *IL18* and markers of various immune cells in the Tumor Immune Estimation Resource (TIMER).

Description	Gene Markers	SKMC				COAD			
		None		Purity		None		Purity	
		Cor	*P*	Cor	*P*	Cor	*P*	Cor	*P*
**CD8^+^ T cell**	*CD8A*	0.748	***	0.573	***	0.166	***	0.114	*
	*CD8B*	0.732	***	0.536	***	0.070	0.047	0.041	0.249
**T cell (general)**	*CD3D*	0.785	***	0.596	***	0.160	***	0.104	*
	*CD3E*	0.772	***	0.572	***	0.123	**	0.056	0.114
	*CD2*	0.805	***	0.639	***	0.218	***	0.168	***
**B cell**	*CD19*	0.547	***	0.334	***	0.108	*	0.048	0.175
	*CD79A*	0.604	***	0.374	***	0.109	*	0.044	0.208
**Monocyte**	*CD86*	0.812	***	0.677	***	0.109	*	0.046	0.188
	*CD115 (CSF1R)*	0.757	***	0.615	***	−0.120	*	−0.193	***
**TAM**	*CCL2*	0.570	***	0.330	***	−0.011	0.744	−0.072	0.041
	*CD68*	0.471	***	0.233	***	−0.048	0.174	−0.100	*
	*IL10*	0.678	***	0.428	***	0.100	*	0.057	0.106
**M1 Macrophage**	*INOS (NOS2)*	0.030	0.524	−0.086	0.065	0.235	***	0.220	***
	*IRF5*	0.623	***	0.378	***	−0.149	***	−0.161	***
	*COX2 (PTGS2)*	0.070	0.134	−0.046	0.325	0.085	0.015	0.053	0.128
**M2 Macrophage**	*CD163*	0.618	***	0.434	***	−0.018	0.605	−0.089	0.011
	*VSIG4*	0.615	***	0.459	***	−0.063	0.071	−0.128	0.000
	*MS4A4A*	0.702	***	0.540	***	0.062	0.076	0.004	0.911
**Neutrophils**	*CD66b (CEACAM8)*	−0.070	0.136	−0.052	0.263	0.087	0.013	0.105	*
	*CD11b (ITGAM)*	0.661	***	0.514	***	−0.114	0.001	−0.194	***
	*CCR7*	0.688	***	0.425	***	0.068	0.053	0.003	0.928
**Natural killer cell**	*KIR2DL1*	0.359	***	0.198	***	0.071	0.043	0.042	0.228
	*KIR2DL3*	0.507	***	0.284	***	−0.013	0.714	−0.046	0.194
	*KIR2DL4*	0.615	***	0.421	***	0.187	***	0.146	***
	*KIR3DL1*	0.475	***	0.264	***	0.068	0.051	0.030	0.388
	*KIR3DL2*	0.565	***	0.340	***	0.075	0.032	0.035	0.322
	*KIR3DL3*	0.211	***	0.149	**	0.064	0.068	0.055	0.117
	*KIR2DS4*	0.442	***	0.295	***	0.061	0.081	0.043	0.217
	*KLRK1 (NKG2D)*	0.697	***	0.516	***	0.198	**	0.138	*
	*NCR1 (NKp46)*	0.526	***	0.362	***	0.159	*	0.095	0.055
	*NCR2 (NKp44)*	0.255	***	0.179	**	0.105	0.024	0.092	0.065
	*NCR3 (NKp30)*	0.713	***	0.503	***	0.133	0.004	0.066	0.187
**Dendritic cell**	*HLA-DPB1*	0.790	***	0.626	***	0.025	0.468	−0.051	0.150
	*HLA-DQB1*	0.725	***	0.537	***	0.068	0.051	0.020	0.573
	*HLA-DRA*	0.808	***	0.658	***	0.148	***	0.093	0.008
	*HLA-DPA1*	0.764	***	0.610	***	0.101	*	0.039	0.269
	*BDCA1 (CD1C)*	0.612	***	0.396	***	0.069	0.048	0.019	0.596
	*BDCA4 (NRP1)*	0.372	***	0.217	***	−0.037	0.286	−0.120	*
	*CD11c (ITGAX)*	0.608	***	0.356	***	−0.049	0.164	−0.132	**
**Th1**	*T-bet (TBX21)*	0.737	***	0.536	***	0.087	0.013	0.026	0.464
	*STAT4*	0.754	***	0.589	***	0.173	***	0.123	0.000
	*STAT1*	0.549	***	0.388	***	0.164	***	0.126	0.000
	*IFNγ (IFNG)*	0.724	***	0.560	***	0.167	***	0.135	0.000
	*TNFα (TNF)*	0.711	***	0.531	***	0.064	0.066	0.027	0.445
**Th2**	*GATA3*	0.833	***	0.686	***	−0.102	0.003	−0.172	***
	*STAT6*	0.007	0.886	−0.026	0.586	−0.079	0.024	−0.069	0.048
	*STAT5A*	0.181	***	0.168	**	−0.141	***	−0.170	***
	*IL13*	0.227	***	0.127	*	0.046	0.186	0.017	0.633
**Tfh**	*BCL6*	0.383	***	0.268	***	−0.054	0.126	−0.121	0.001
	*IL21*	0.487	***	0.348	***	0.008	0.809	−0.013	0.711
**Th17**	*STAT3*	0.300	***	0.211	***	0.065	0.065	0.026	0.459
	*IL17A*	−0.009	0.856	−0.091	0.051	0.099	*	0.100	0.004
**Treg**	*FOXP3*	0.690	***	0.462	***	−0.076	0.029	−0.152	***
	*CCR8*	0.682	***	0.513	***	−0.035	0.314	−0.096	0.006
	*STAT5B*	0.226	***	0.249	***	−0.323	***	−0.329	***
	*TGF* *β (TGFB1)*	0.431	***	0.207	***	−0.024	0.486	−0.103	0.003
**T cell exhaustion**	*PD1 (PDCD1)*	0.740	***	0.555	***	0.078	0.026	0.019	0.586
	*CTLA4*	0.454	***	0.199	***	0.062	0.079	−0.002	0.950
	*LAG3*	0.703	***	0.512	***	0.094	0.008	0.037	0.290
	*TIM3 (HAVCR2)*	0.777	***	0.610	***	0.084	0.016	0.021	0.543

SKCM, skin cutaneous melanoma; COAD, colon adenocarcinoma; TAM, tumor-associated macrophage; Th, T helper cell; Tfh, follicular helper T cell; Treg, regulatory T cell; Cor, R value of Spearman’s correlation; None, correlation without adjustment. Purity, correlation adjusted by purity. * *P* < 0.01; ** *P* < 0.001; *** *P* < 0.0001.

**Table 2 jcm-08-01993-t002:** Correlation between *IL18* and markers of CD8^+^ T and natural killer (NK) cells in GEPIA.

Cell Type	Gene Markers	SKCM				COAD			
		Tumor		Normal		Tumor		Normal	
		*R*	*P*	*R*	*P*	*R*	*P*	*R*	*P*
**CD8^+^ T cells**	*CD8A*	0.63	***	−0.11	*	−0.11	0.067	0.41	***
	*CD8B*	0.63	***	−0.13	*	−0.008	0.089	0.38	***
**NK cells**	*KIR2DL1*	0.22	***	−0.091	0.031	0.015	0.8	0.1	0.053
	*KIR2DL3*	0.3	***	−0.07	0.097	0.006	0.92	0.21	***
	*KIR2DL4*	0.44	***	−0.075	0.079	0.11	0.064	0.66	***
	*KIR3DL1*	0.25	***	−0.1	0.013	0.087	0.15	0.24	***
	*KIR3DL2*	0.42	***	−0.11	*	0.043	0.48	0.43	***
	*KIR3DL3*	0.19	***	0.06	0.16	0.032	0.6	0.12	0.02
	*KIR2DS4*	0.16	**	−0.16	**	0.015	0.8	0.18	**
	*KLRK1 (NKG2D)*	0.63	***	−0.11	*	0.094	0.12	0.45	***
	*NCR1 (NKp46)*	0.29	***	−0.086	0.043	0.081	0.19	0.15	*
	*NCR2 (NKp44)*	0.075	0.11	−0.018	0.67	0.21	**	0.38	***
	*NCR3 (NKp30)*	0.46	***	−0.11	0.011	0.083	0.17	0.26	***

* *P* < 0.01; ** *P* < 0.001; *** *P* < 0.0001.

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
