# Peer review of "Interleukin-18 Is a Prognostic Biomarker Correlated with CD8+ T Cell and Natural Killer Cell Infiltration in Skin Cutaneous Melanoma"

_jcm, 2019, doi:10.3390/jcm8111993_

Round 1
Reviewer 1 Report
In the current study, the authors demonstrate a lower mRNA expression level of IL-18 in SKCM compared with normal tissues. The authors further characterize high IL-18 expression level is a favorable indicator for overall survival of SKCM, potentially through regulating immune infiltration where CD8+ T and NK cells function as anti-tumor effectors. Overall the results present here are interesting and clinical significant. But I have some concerns as below:
Figure 2a. Clinicopathological characteristics of the cohorts used should be briefly described. Did the author adjust/normalize the data based on patient age, gender, stages of melanoma when performing survival analysis?
Figure 2. How did the author setup cutoff value/threshold to define “high” and “low” IL-18 expression level? Not sure it is scientifically reasonable to arbitrarily separate cases into two groups only by percentage (50% and 50%), if this is what the authors have performed.
Line 187, I assume it should be Figure 3b. Line 195, I assume it should E192K. Figure 3b, how many patients in each group? How the numbers on Y axis were normalized?
Figure 3e, it is hard to ignore the 3.31% amplifications of IL-18 gene among all alterations. The authors should be very careful to make the statement in Line 202 just according to the current Figure 3e and 3f.
Figure 4. It is impossible to read the numbers in each panel.
Several studies have shown that IL-18 plays a critical role in the initiation of metastasis of melanoma. High expression of IL-18 correlates with immune cells recruitment and infiltration as shown in this study, but may also trigger the adhesion to endothelium cells as well as activate different pathways that contribute to melanoma cell proliferation and migration. Can author provide the explanation of the potential discrepancy?
Author Response
Nov 09, 2019
Manuscript ID: jcm-635449
Title: Interleukin-18 is a prognostic biomarker correlated with CD8+ T cell and natural killer cell infiltration in skin cutaneous melanoma.
Dear editor,
Please consider our revised manuscript entitled “Interleukin-18 is a prognostic biomarker correlated with CD8+ T cell and natural killer cell infiltration in skin cutaneous melanoma.” for publication. We wish to thank you and the reviewers for the grateful comments. We carefully considered and addressed the reviewer’s comments, as presented (red colored) like below.
Many thanks again for the reviewers' valuable and constructive comments. Our manuscript was already checked by a professional English editing company (Editage). We really hope that the revised manuscript and our responses to all the reviewers’ precious comments will be satisfactory to the editor and the reviewers.
Thanks again for your kindness.
Sincerely yours
Kyung Eun Kim, Ph.D.
Responses to the Reviewers’ Comments
Reviewer #1:
Comment: In the current study, the authors demonstrate a lower mRNA expression level of IL-18 in SKCM compared with normal tissues. The authors further characterize high IL-18 expression level is a favorable indicator for overall survival of SKCM, potentially through regulating immune infiltration where CD8+ T and NK cells function as anti-tumor effectors. Overall the results present here are interesting and clinical significant. But I have some concerns as below:
Response: Thank you for your thoughtful comments. We carefully considered and addressed your comments as presented like below.
Comment #1: Figure 2a. Clinicopathological characteristics of the cohorts used should be briefly described. Did the author adjust/normalize the data based on patient age, gender, stages of melanoma when performing survival analysis?
Response #1: Thank you for your comments. All data in Figure 2 uses TCGA datasets. The number of patients with each clinicopathological characteristic in TCGA subsets are available at previous published article and described it with citation in Experimental section briefly (Line 104~105). As your suggestion, we tried to normalize the data, however, we could not adjust/normalized the data based on patient clinicopathological characteristic because GEPIA performs survival analysis based on gene expression only for the total data from specific cancer-type.
<References>
1. Liu J et al. An Integrated TCGA Pan-Cancer Clinical Data Resource to Drive High-Quality Survival Outcome Analytics. Cell. 2018 Apr 5;173(2):400-416.
Comment #2: Figure 2. How did the author setup cutoff value/threshold to define “high” and “low” IL-18 expression level? Not sure it is scientifically reasonable to arbitrarily separate cases into two groups only by percentage (50% and 50%), if this is what the authors have performed.
Response #2: We perform overall survival analysis using GEPIA with “median” group cutoff option, which means groups were separated by median value of gene expression (50% high gene expression and 50% low
gene expression). Group cutoff value and proportion of two groups could be further optimized to minimize p-value. However, significant difference in survival between 50% high gene expression and 50% low gene expression groups could be acceptable as supportive evidence of the correlation of specific gene expression on the patient survival as shown in many other previous articles [1-3].
We revised section 2.2 in experimental section to explain it in detail (Line 98~103).
<References>
2. Sun C et al. Elevated DSN1 expression is associated with poor survival in patients with hepatocellular carcinoma. Hum Pathol. 2018 Nov;81:113-120.
3. Yuan L et al. A novel correlation between ATP5A1 gene expression and progression of human clear cell renal cell carcinoma identified by co‑expression analysis. Oncol Rep. 2018 Feb;39(2):525-536.
4. Zhang YL et al. High Expression B3GAT3 Is Related with Poor Prognosis of Liver Cancer. Open Med (Wars). 2019 Feb 26;14:251-258.
Comment #3: Line 187, I assume it should be Figure 3b. Line 195, I assume it should E192K. Figure 3b, how many patients in each group? How the numbers on Y axis were normalized?
Response #3: Thank you for the comments. We changed it as you suggested (Line 198, Line 199, Line 206). We appreciate your help.
To generate Figure 3b, we utilized SurvExpress biomarker validation tool which uses log2 quantile-normalized intensity as expression value from datasets [1]. We changed the label and add a sentence in figure legend to clarify it as shown below (Line 220~221).
<References>
1. Aguirre-Gamboa R et al. SurvExpress: an online biomarker validation tool and database for cancer gene expression data using survival analysis. PLoS One. 2013 Sep 16;8(9):e74250.
Comment #4: Figure 3e, it is hard to ignore the 3.31% amplifications of IL-18 gene among all alterations. The authors should be very careful to make the statement in Line 202 just according to the current Figure 3e and 3f.
Response #4: We regret the improper explanation specially in Figure3 legend that could cause the confusion. “mRNA High” in Figure 3e represents the frequencies of samples that has bigger than 2 of the mRNA Expression Zscores, RSEM (Batch normalized from Illumina HiSeq_RNASeqV2). Therefore, “mRNA high” is not the “amplification” CNA type of IL18 gene identified by GISTIC algorism. Actually, as shown in Figure 3f, there were no cases that has amplification of IL18 gene. To avoid confusion, we opted out the mRNA expression data from Figure 3e and only focus on the mutation and CNA in Figure 3d-f. We rewrote the figure legends for 3d-f and results section according to the corresponding change (Line 205~207, Line 211~213, Line 226~231).
Comment #5: Figure 4. It is impossible to read the numbers in each panel.
Response #5: We edited the figures (Figure 3, 4, 5, and 6) and changed into higher resolution (300dpi-> 600dpi) picture to enhance the readability of characters. We also added Supplementary Table S5 to show the correlation constants and p-values written on each diagram.
Comment #6: Several studies have shown that IL-18 plays a critical role in the initiation of metastasis of melanoma. High expression of IL-18 correlates with immune cells recruitment and infiltration as shown in
this study, but may also trigger the adhesion to endothelium cells as well as activate different pathways that contribute to melanoma cell proliferation and migration. Can author provide the explanation of the potential discrepancy?
Response #6: As you are concerned, previous studies have shown that recombinant IL-18 directly promotes the growth and metastasis of B16F10 murine melanoma cell line [1]. In addition, IL-18 is an inflammatory cytokine and has been reported to show a pro-cancer effect by being involved in inflammatory reactions such as adhesion to endothelial cells and angiogenesis [2].
(A) However, these in vitro studies have limited biological and clinical importance compared to the clinical observation like IL-18 expression in cancer tissue. To date, there are no articles that have been analyzed and reported comprehensively about the effect of IL-18 expression on the clinical outcome in malignant melanoma in humans. Therefore, this study is considered to be the first report to have clinical significance.
(B) Such in vitro studies usually cannot reflect the actual tumor microenvironment in which various other regulatory factors could regulate cancer cells synergistically or antagonistically with IL-18.
For example, a lot of studies have reported that IL-18 shows anti-cancer effects in combination with other cytokines such as IL-12 and IL-15, as results of increased T cell proliferation and NK cell activity [3, 4]. Indeed, overexpression of IL-18 in a mouse model of malignant melanoma has been shown to increase tumor cell apoptosis, inhibit tumor growth, reduce lung metastasis, and inhibit angiogenesis. The effect is synergistic when used in combination with dacarbazine, a treatment for malignant melanoma, showing IL-18 shows systemically a significant anti-cancer effect in the body [5]. Mirjačić Martinović K et al. reported that NK cells isolated form patient with malignant melanoma exhibited increased NK cell activity when treated with IL-18 and IL-12, whereas IL-18 alone did not show those effect [6], and Müller J et al. reported that IL-18 and IL-12 combination gene therapy showed significant tumor regression in a malignant melanoma-induced horse model [7].
In addition, several studies reported the importance of doses of IL-18. They proved that the anti-cancer effects of IL-18 involving the activation of NK cells and T cells are shown with high doses of IL-18 often combined with IL-2 or IL-12 [8-10]. These results are consistent with our data, which indicate that higher expression of IL-18 has a better prognosis.
Collectively, these results suggest that the IL-18 might show both anti-cancer and pro-cancer effects depending on tumor environment [11]. Therefore, further investigations are needed about the contradictory findings related to the biological role of IL-18 in vitro and in vivo.
We revised the discussion to explain the potential discrepancy as shown in Line 356~386.
<References>
1. Jung MK et al. IL-18 enhances the migration ability of murine melanoma cells through the generation of ROI and the MAPK pathway. Immunol Lett. 2006 Nov 15;107(2):125-30.
2. Valcárcel M et al. IL-18 regulates melanoma VLA-4 integrin activation through a Hierarchized sequence of inflammatory factors. J Invest Dermatol. 2014 Feb;134(2):470-480.
3. Ni J et al. Sustained effector function of IL-12/15/18-preactivated NK cells against established tumors. J Exp Med. 2012 Dec 17;209(13):2351-65.
4. Choi IK et al. Oncolytic adenovirus co-expressing IL-12 and IL-18 improves tumor-specific immunity via differentiation of T cells expressing IL-12Rβ2 or IL-18Rα. Gene Ther. 2011 Sep;18(9):898-909.
5. Yang C et al. Oncolytic adenovirus expressing interleukin-18 improves antitumor activity of dacarbazine for malignant melanoma. Drug Des Devel Ther. 2016 Nov 15;10:3755-3761.
6. Mirjačić Martinović K et al. Favorable in vitro effects of combined IL-12 and IL-18 treatment on NK cell cytotoxicity and CD25 receptor expression in metastatic melanoma patients. J Transl Med. 2015 Apr 14;13:120.
7. Müller J et al. Double-blind placebo-controlled study with interleukin-18 and interleukin-12-encoding plasmid DNA shows antitumor effect in metastatic melanoma in gray horses. J Immunother. 2011 Jan;34(1):58-64.
8. Dinarello CA. Interleukin-18. Methods. 1999 Sep;19(1):121-32.
9. Son YI et al. Interleukin-18 (IL-18) synergizes with IL-2 to enhance cytotoxicity, interferon-gamma production, and expansion of natural killer cells. Cancer Res. 2001 Feb 1;61(3):884-8.
10. Coughlin CM et al. Interleukin-12 and interleukin-18 synergistically induce murine tumor regression which involves inhibition of angiogenesis. J Clin Invest. 1998 Mar 15;101(6):1441-52.
11. Lebel-Binay S et al. Interleukin-18: biological properties and clinical implications. Eur Cytokine Netw. 2000 Mar;11(1):15-26.

Reviewer 2 Report
The manuscript submitted by Gil and Kim is the systems analysis of Tumor-Infiltrating Lymphocytes (TIL) and their functional regulation through the IL-18 gene. In this exciting, well-defined and extensive analysis using the only public databases, authors have shown that the skin cutaneous melanoma progression can be predicted by measuring IL-18, perforin and granzyme gene expression in CD8 and NK cells associated with patients. This manuscript deserves to be published in the Journal of Clinical Medicine after addressing minor suggestions.
Authors are suggested to refer to the landscape study published in Nature Medicine 2016 by Gentle et al(PMID: 26193342). This study has analyzed a larger number of TIL and its prognostic significance. As it can be realized that the strategy to use databases, an algorithm can have a profound impact on the results and observation. For example, the authors should comment on the use of CIBERSORT, missing gd T cells which have better prognostic significance and anti-tumor properties, lack of observations on NKG2D receptor-ligand genes, PD1-PDL1 genes as they are also known to contribute to TIL function and prognostic significance, missing RoRgt in Table 1.
In such extensive studies based on public databases, computational/bioinformatics, and statistical analysis, authors are suggested to include detail methods for given results including statistical significance cut-off (for example, section 3.2). Also, mention how the IL-18 gene was defined as high and low in the patient cohort represented in figure 2. Another important aspect needs to mention in the methods section is about normalization approaches. Usually, such systems studies are very sensitive to false positive/negative results due to poor or no use of normalization methods (for example, results represented in figure 1). The SurvExpress biomarker validation tool should be mentioned in the methods section.
Could it be possible to mention why authors have used p-value instead of q-value (or adjusted p-value or FDR)? Some tests like GZMB (cor=0.52, p-value=3.03e-77) makes it difficult to believe such a great statistical significance.
An additional supplementary table on the list of GEO databases used in this study will help readers to understand the results (for example in section 3.3).
In conclusion, the authors should comment on an outlook or future steps to the current study.
Please check the typos in the text. Line 187 figure 3a should be written as 3b, line 195 the mutation E152K should be written as E192K as mentioned in the figure.
Author Response
Nov 09, 2019
Manuscript ID: jcm-635449
Title: Interleukin-18 is a prognostic biomarker correlated with CD8+ T cell and natural killer cell infiltration in skin cutaneous melanoma.
Dear editor,
Please consider our revised manuscript entitled “Interleukin-18 is a prognostic biomarker correlated with CD8+ T cell and natural killer cell infiltration in skin cutaneous melanoma.” for publication. We wish to thank you and the reviewers for the grateful comments. We carefully considered and addressed the reviewer’s comments, as presented (red colored) like below.
Many thanks again for the reviewers' valuable and constructive comments. Our manuscript was already checked by a professional English editing company (Editage). We really hope that the revised manuscript and our responses to all the reviewers’ precious comments will be satisfactory to the editor and the reviewers.
Thanks again for your kindness.
Sincerely yours
Kyung Eun Kim, Ph.D.
Responses to the Reviewers’ Comments
Reviewer #2:
Comment: The manuscript submitted by Gil and Kim is the systems analysis of Tumor-Infiltrating Lymphocytes (TIL) and their functional regulation through the IL-18 gene. In this exciting, well-defined and extensive analysis using the only public databases, authors have shown that the skin cutaneous melanoma progression can be predicted by measuring IL-18, perforin and granzyme gene expression in CD8 and NK cells associated with patients. This manuscript deserves to be published in the Journal of Clinical Medicine after addressing minor suggestions. Response: Thank you for your thoughtful comments. We carefully considered and addressed your comments as presented like below.
Comment #1: Authors are suggested to refer to the landscape study published in Nature Medicine 2016 by Gentle et al (PMID: 26193342). This study has analyzed a larger number of TIL and its prognostic significance. As it can be realized that the strategy to use databases, an algorithm can have a profound impact on the results and observation. For example, the authors should comment on the use of CIBERSORT, missing gd T cells which have better prognostic significance and anti-tumor properties, lack of observations on NKG2D receptor-ligand genes, PD1-PDL1 genes as they are also known to contribute to TIL function and prognostic significance, missing RoRgt in Table 1.
Response #1: Thank you for your thoughtful comments.
(1) According to Nature Medicine 2016 by Gentle AJ et al., T cells as well as CD8+ and NK cells are important indicators of favorable outcomes in cancer [1]. Also, T cells are known for their ability to secrete cytokines such as IL-18 and IL-12 [2]. Therefore, as your suggestion, we added a correlation between IL18 expression and T cell-related gene expression. For this analysis, we examined transcriptomic data of two genes (CCR5 and CXCR6) from LM22 gene set along with RORγt and CD27. Newman AM et al. reported significantly differentially expressed genes in T cells by comparing each 22 leukocyte subsets [3; SuppTable1_DEGs]. However, those genes are not specific markers for only T cells. It is well known that the genes also expresses in other immune cells. Therefore, we added the correlation data as supplementary data and mentioned the limitation of current study in the discussion part (Line 422~442). Also, the results of four of these genes that are highly correlated with IL-18 have been added to Supplementary Table S6~S9.
(2) In particular, the results of the paper using CIBERSORT show that the higher the level of CD8+ T cell [4] or NK cell [5] in SKCM is positively correlated with a better prognosis. These studies suggest that increased infiltration of T cells, CD8+ T cells, and NK cells is an important target for estimating the prognosis of cancer patients with SKCM. We mentioned the importance of T cells as TILs, and revised the result (Line 284~285, Line 289~299) and discussion section (Line 398~400, Line 411~421).
(3) As you suggest, we have added the correlation of IL18 expression with various NK cells activating receptor genes as shown in Table 1 and 2.
(4) In addition, this study provided clinical significance that IL18 expression could be a good prognostic indicator in patients with SKCM, however, it had a positive correlation with the expression of immune suppressive molecules such as PD-1 and CTLA4 as shown in Table 1. In fact, studies have shown that IL-18 increases PD-1 expression, leading to immunosuppression in NK cells in breast cancer and melanoma [6, 7]. Recently, anti-PD-1/anti-PD-L1 therapy has been successful in treating cancer, but its effectiveness is limited by the infiltration of immune cells and the characteristics of cancer [8, 9]. Despite the increased expression of immune suppressive molecules, this study found that SKCM patients with higher IL18 expression showed overall higher survival rates. Therefore, we suggest the need for additional analysis to confirm the efficacy of anti-PD-1 therapeutics according to differences in IL18 expression (Line 443~451).
<References>
1. Gentles AJ et al. The prognostic landscape of genes and infiltrating immune cells across human cancers. Nat Med. 2015 Aug;21(8):938-945.
2. Fergusson JR et al. CD161 defines a transcriptional and functional phenotype across distinct human T cell lineages. Cell Rep. 2014 Nov 6;9(3):1075-88.
3. Newman AM et al. Robust enumeration of cell subsets from tissue expression profiles. Nat Methods. 2015 May;12(5):453-7.
4. Chen B et al. Profiling Tumor Infiltrating Immune Cells with CIBERSORT. Methods Mol Biol. 2018;1711:243-259.
5. Cursons J et al. A Gene Signature Predicting Natural Killer Cell Infiltration and Improved Survival in Melanoma Patients. Cancer Immunol Res. 2019 Jul;7(7):1162-1174.
6. Park IH et al. Tumor-derived IL-18 induces PD-1 expression on immunosuppressive NK cells in triple-negative breast cancer. Oncotarget. 2017 May 16;8(20):32722-32730.
7. Terme M et al. IL-18 induces PD-1-dependent immunosuppression in cancer. Cancer Res. 2011 Aug 15;71(16):5393-9.
8. Herbst RS et al. Predictive correlates of response to the anti-PD-L1 antibody MPDL3280A in cancer patients. Nature. 2014 Nov 27;515(7528):563-7.
9. Guirgis HM. The impact of PD-L1 on survival and value of the immune check point inhibitors in non-small-cell lung cancer; proposal, policies and perspective. J Immunother Cancer. 2018 Feb 20;6(1):15.
Comment #2: In such extensive studies based on public databases, computational/bioinformatics, and statistical analysis, authors are suggested to include detail methods for given results including statistical significance cut-off (for example, section 3.2). Also, mention how the IL-18 gene was defined as high and low in the patient cohort represented in figure 2. Another important aspect needs to mention in the methods section is about normalization approaches. Usually, such systems studies are very sensitive to false positive/negative results due to poor or no use of normalization methods (for example, results represented in figure 1). The SurvExpress biomarker validation tool should be mentioned in the methods section.
Response #2: (1) Thank you for your comments. We added the statistically significant cut-off value (p < 0.05) in result section 3.2 and experimental section 2.2 . We also added the detailed explanation how high IL18 expression and low IL18 expression groups were divided in section 2.2 (Line 98~104, 168,175,176).
(2) We added the normalization methods of gene expression data in GEPIA and GENT database with references in section 1.1 for the figure 1 (Line 86~92).
(3) We already mentioned the SurvExpress biomarker validation tools in section 2.2 of Experimental sections (Line 105~107).
Comment #3: Could it be possible to mention why authors have used p-value instead of q-value (or adjusted p-value or FDR)? Some tests like GZMB (cor=0.52, p-value=3.03e-77) makes it difficult to believe such a great statistical significance.
Response #3: We utilized TIMER web tool to examine the correlation of IL-18 and GZMB and PRF1 in this figure. TIMER webtool use “Spearman’s correlation analysis” and only provides the p-value for the significance determination.
(A) The q-value is defined as the minimum false discovery rate at which an observed score is deemed significant. The q-value is an analog of the p-value that incorporates multiple testing correction. For any genome-wide analysis, reporting individual p-values can be misleading, because the p-value does not correct for the large number of tests (each gene expression data set is considered as one test) Instead, q-value is useful. However, in correlation study could not be considered as multiple testing because it only uses two genes data.
(B) The p-values were provided by TIMER databases. If I used different tools to carried out spearman’s correlation analysis with SKCM-TCGA datasets, we could get similarly lower level of p-values (GEPIA : R = 0.68, p−value = 1.3e−62) Regardless of tools used, great statistical significance between the gene expression of IL-18 and GZMB was identified.
Comment #4: An additional supplementary table on the list of GEO databases used in this study will help readers to understand the results (for example in section 3.3). Response #4: We added the dataset information in Supplementary Table S4. Figure 3b and 3c were generated from same geo dataset (GSE19234) but processed by different database tools.
Comment #5: In conclusion, the authors should comment on an outlook or future steps to the current study.
Response #5: As you suggested, we revised the discussion to suggest the need for further studies as shown in Line 366~387 and Line 423~452.
Comment #6: Please check the typos in the text. Line 187 figure 3a should be written as 3b, line 195 the mutation E152K should be written as E192K as mentioned in the figure.
Response #6: Thank you for the comments. We corrected typos as shown in Line 198 and Line 206..
